# Do Fungicides Affect Alkaloid Production in *Catharanthus roseus* (L.) G. Don. Seedlings?

**DOI:** 10.3390/molecules28031405

**Published:** 2023-02-01

**Authors:** Purin Candra Purnama, Leonardo Castellanos Hernandez, Robert Verpoorte

**Affiliations:** 1Department of Biology, Faculty of Mathematics and Natural Science, Universitas Sebelas Maret, Jl. Ir. Sutami 36A, Surakarta 57126, Indonesia; 2Natural Product Laboratory, Institute of Biology, Leiden University, Sylviusweg 72, 2333 AG Leiden, The Netherlands; 3Marine Natural Products Group, Department of Chemistry, Universidad Nacional De Colombia, Carrera 45 # 26-85 Edif. Uriel Gutiérrez, Bogota 110110, Colombia

**Keywords:** *Catharanthus roseus*, fungicide, ^1^HNMR, HPLC, alkaloids

## Abstract

The presence of endophytes in plants is undeniable, but how significant their involvement is in the host plant biosynthetic pathways is still unclear. The results reported from fungicide treatments in plants varied. Fungicide treatment in *Taxus* was found to decrease the taxol content. In *Ipomoea asarifolia*, Pronto Plus and Folicur treatments coincided with the disappearance of ergot alkaloids from the plant. In *Narcissus pseudonarcissus* cv. Carlton, a mixture of fungicide applications decreased the alkaloids concentration and altered the carbohydrate metabolism. *Jacobaea* plants treated with Folicur reduced the pyrrolizidine alkaloids content. There have not been any studies into the involvement of endophytic fungi on alkaloids production of *Catharanthus roseus* until now. Though there is a report on the isolation of the endophytic fungi, *Fusarium oxysporum* from *C. roseus*, which was reported to produce vinblastine and vincristine in vitro. To detect possible collaborations between these two different organisms, fungicides were applied to suppress the endophytic fungi in seedlings and then measure the metabolomes by ^1^HNMR and HPLC analysis. The results indicate that endophytic fungi were not directly involved in alkaloids biosynthesis. Treatment with fungicides influenced both the primary and secondary metabolism of *C. roseus.* The systemic fungicides Pronto Plus and Folicur caused an increase in loganin and secologanin levels. In contrast, control samples had higher level of catharanthine and vindoline. This means that fungicide treatments cause changes in plant secondary metabolism.

## 1. Introduction

The fact that all plants harbor diverse spectra of microorganisms is widely accepted nowadays. As reviewed by Berendsen and colleagues [1] and also Vorholt [2], plant’s rhizosphere and phyllosphere are rich in microorganisms. The other plant’s part that can be inhabited by microorganisms is its internal environment in which interactions are found that range from pathogenism to endophytism [3].

The term endophyte was first mentioned in 1866 by De Bary [4] to define all organisms that colonize plant tissues. Several definitions have been proposed. The most recent says that endophytes include all microorganisms that complete their full life cycles inside the plant tissues without causing apparent symptoms. This last definition is probably the most practical for molecular plant biologists because it includes the various symptomless fungi likely to contaminate plant genomic DNA extracts [5,6,7]. As mentioned by Petrini, if each plant species has (at least) one specific endophyte, there must be 300,000 endophytes species [4]. Accordingly, endophytes have significant biodiversity waiting for further exploration, among others related to their use as an alternative source of natural products, as has been reviewed by Strobel and Daisy [8], Berdy [9], Zhang and colleagues [6], Gunatilaka [10], Ludwig-Mueller [11], and Daley and Cordell [12].

The discovery of a taxol-producing endophytic fungi, *Taxomyces andreanae*, isolated from the inner bark of *Taxus* sp. tree by Stierle and colleagues [13] initiated similar work on other plants [13,14,15,16]. However, the underlying mechanisms of how endophytic fungi could produce the same very complex metabolites as the host plant are not yet known.

In general, there are two different types of fungicides based on their mode of action, systemic (like Pronto Plus and Folicur) and non-systemic fungicides (like Switch). The active constituent of Pronto Plus (Bayer-CropScience, Leverkussen, Germany) are tebuconazole 133g/l and spiroxamine 250g/l, whilst Folicur (Bayer-CropScience, Leverkussen, Germany) only has tebuconazole 251 g/l as an active constituent. The active constituents in Switch (Syngenta Agro, Frankfurt AM Germany) are 375 g/l cyprodinil and 250 g/l fludioxonil. All the compounds in systemic fungicides are absorbed and distributed through xylem into the entirety of the plant. In non-systemic fungicides, the compounds act locally, at the site of application only (www.frac.info, accessed on 27 October 2014).

Lubbe and co-workers [17] applied a mixture of fungicides in the field to leaves of *Narcissus pseudonarcissus* cv. Carlton. The alkaloid concentrations decreased and the carbohydrate metabolism was altered. Soliman and coworkers [18] found that fungicide treatments of intact *Taxus* trees decrease the taxol content, together with reduced levels of the proteins, that correspond to two critical steps required for taxol biosynthesis. Systemic fungicides’ (Pronto Plus and Folicur) treatments of *Ipomoea asarifolia* eliminated the endophyte *Periglandula ipomoeae*, which coincided with the disappearance of ergot alkaloids from the plant [19,20,21]. Wilson and colleagues [22] discovered that endophytes residing in *Hordeum brevisubulatum* subsp. *violaceum* were reduced after treatment with propiconazole (systemic fungicide) instead of prochloraz (non-systemic fungicide). In contrast, an increasing growth of endophytes was also observed after the application of two non-systemic fungicides, terrazole and chloroneb [23]. A lower concentration of pyrrolizidine alkaloids (PAs) was detected in *Jacobaea* plants treated with systemic fungicide (Folicur) and an endophyte of the Glomus type became detectable in the plant by PCR using β-tubulin and ITS primers [24].

Apparently, each type of fungicide has its own specific effect on different endophytic fungi, which is in line with the general knowledge of the specificity of fungicides for certain classes of fungi. Although it seems that fungicide treatment influences a plant’s metabolome, there are only a few reports on the influence of fungicides on endophytic fungi in crop plants.

Related to *C. roseus,* thus far there have only been reports on the regrowth of 183 endophytic fungi and an antibiotic-producing *Streptomyces* sp. [25]. Surprisingly, in 2013, Kumar and coworkers [26] reported the isolation of vinblastine and vincristine from *Fusarium oxysporum* grown in vitro. Unfortunately, no effort was made to prove the presence of some of the many known genes of the biosynthesis of these complex alkaloids that involve at least four different cell types in the plant leaves. Neither were any of the other major alkaloids found.

There are no reports, thus far, on the influence of fungicide application on *C. roseus* plants. The aim of our study was to explore a possible role of endophytic fungi in the biosynthesis of alkaloids in *Catharanthus* plants by measuring alkaloids’ contents after different fungicide treatments. In addition, we also investigated possible other effects of the fungicide treatments on the plants metabolism by using a metabolomics approach. The large amount of data generated by ^1^HNMR spectroscopy were analyzed by both non-supervised and supervised multivariate data analysis (MvDA).

## 2. Results

### 2.1. Optimization of Various Fungicides’ Concentrations on Chaetomium sp. Growth

*Chaetomium* sp. only grew at 0.001% Folicur and not in the other fungicides at this concentration. Therefore, the concentrations applied for further experiments were 0.001% for Switch and Pronto Plus and 0.005% for Folicur. Those concentrations were considered as effective to kill a fungus without affecting the physiological state of the plants.

### 2.2. Dry Weight

The dry weight between the control and fungicides-treated seedlings (Figure 1) showed significant difference (ANOVA, F (3,44) = 5.387, *p*= 0.003). Control had the highest dry mass compared to all treatments. Among harvesting days, there was also a significant difference observed (ANOVA, F (3,44) = 6.928, *p*= 0.001), with 28 days having the highest dry weight.

### 2.3. Alkaloids Accumulation

Three compounds were identified by HPLC, loganin—the precursor of TIAs biosynthesis—and TIAs catharanthine and vindoline (Figure 2). No strictosidine could be found. Loganin (ANOVA, F (3,44) = 3.938, *p* = 0.014) in the control was present at a level more than 10-fold higher than the alkaloids. With Pronto Plus and Folicur, a considerable increase of the loganin content was observed. Meanwhile, with Switch there were no significant changes in loganin content if compared to controls.

The Switch, Pronto Plus- and Folicur-treated seedlings showed clearly different patterns of vindoline and catharanthine levels if compared to the controls. Though most differences were not statistically significant. But the increase of catharanthine levels in seedlings at day 21 and 28 (ANOVA, F (3,44) = 3.106, *p* = 0.036) are statistically significantly higher in the controls than in the treated seedlings. The same applies for vindoline, for which the lower levels of the alkaloids seem even more pronounced than in case of catharanthine. It seems that the increase of alkaloids levels in the controls during the 28 days of the experiment is suppressed in the treated seedlings.

### 2.4. ^1^HNMR Measurements

#### 2.4.1. Metabolite Identification

^1^HNMR was used to obtain an overview of the total of metabolites and see the possible effect of the different fungicide treatments on the metabolism of the *C. roseus* seedlings. The signal assignments for various primary and secondary metabolites were performed following previous literature [27] and the in-house metabolites database (see Table 1).

By observing ^1^HNMR spectra of seedlings treated with 0.03% Pronto Plus, we were able to identify a residue of the fungicide in the samples (Figure 3). The signals at δ_HA_ 8.54 (s), δ_HB_ 8.10 (s), δ_HC_ 7.26 (d, *J* = 8.4 Hz), δ_HD_ 7.07 (d, *J* = 8.4 Hz), δ_HF_ 4.52 (d, *J* = 14.7 Hz), δ_HG_ 4.44 (d, *J* = 14.7 Hz), and δ_HE_1.03 (s) were assigned for tebuconazole residue in Pronto Plus-treated samples. By comparing ^1^HNMR spectra between seedlings treated with 0.03% Pronto Plus (red spectra) and 0.001% Pronto Plus (blue spectra), we were able to confirm that there were no fungicides traces detectable in the samples of the low-level treatments (Figure 4).

#### 2.4.2. Multivariate Data Analysis

^1^HNMR metabolomics has the advantage of direct quantitation as the signal intensity of a proton is only dependent on molar concentration, and thus all compounds observed can be directly quantified. Each compound has a unique ^1^HNMR spectra, and the ^1^HNMR spectra of an extract is thus the sum of all the spectra of the compounds present. At the same time, this is a weakness, as a signal overlap might hamper identification and quantitation. Various 2D-NMR spectral method can help to overcome this problem.

To be able to deal with the enormous amount of information, various chemometric methods were used. Most common for ^1^HNMR metabolomics is multivariate data analysis (MvDA). Unsupervised MvDA, such as principal component analysis (PCA), showed in two or three dimensions (score plots) the maximum separation between all samples.

When certain grouping is observed, one could learn from loadings plots what signals make the difference between the groups. The shift of the signals gave information on the identity of the compound involved in the grouping. When many variables are involved, PCA might not show any useful grouping. In that case, supervised MvDA could be applied in which one poses a question, for example giving the maximum separation between two classes of samples, such as the control and fungicides-treated plant. Only signals that separate these two classes would be shown.

Applying PCA to the samples results in Figure 5, this PCA score plot with PC1 and PC2 explained 69% of the variation in all samples. Although a clear trend is observed in which the different time points cluster more from the right upper quadrant to the left lower quadrant, there is considerable overlap. Controls were separated from the treated samples, but between treatments the separation was not so clear. It seemed that after 28 days, all samples started to cluster again. The effect of the fungicides might become less, which might be due to catabolism. To observe differences specifically due to the treatments, supervised MvDA was applied. In Figure 5b, one can see that treatments and controls were separated by OPLS-DA.

The OPLS-DA analysis showed a clear separation between the control and fungicides-treated samples (Figure 5b). Interestingly, the model could also discriminate treatments, especially between systemic (Pronto Plus and Folicur) and non-systemic (Switch) fungicide samples. The model resulted in a variance of R^2^ of 0.951 and a predictive ability Q^2^ of 0.899. The cross validation of the model using CV-ANOVA provided highly significant results (F = 10.3017, *p* < 0.001). The loadings plot of the OPLS-DA (Figure 5c) shows that high signals of loganin and secologanin are typical for distinguishing the Pronto Plus- and Folicur-treated samples from the others, while high signals of sucrose and glucose are observed for the Switch-treated samples (Figure 5c). Controls were distinguished by the signals of catharanthine, vindoline, and 4-O-Caffeoyl quinic acid.

Unsupervised MvDA was also conducted to observe the other complete set of experiments with high concentrations of fungicides at 7 days after: 0.03% of Pronto Plus, 0.03% Folicur, and 0.03% Switch (Figure 6a). The PCA managed to separate 0.03% Pronto Plus from the rest of the samples, such as Folicur, Switch, and control. The loadings scatter plot showed that separation was mostly due to the appearance of the tebuconazole residues (Figure 3). To obtain an insight into the influence of high and low fungicide concentrations on seedlings metabolism, unsupervised MvDA was applied. All the samples were separated into two big groups based on their concentrations. With 0.03% Pronto Plus, seedlings were separated from the rest of the samples. However, the loadings scatter plot shows that discrimination was mainly due to the tebuconazole residue, contributing to the separation (Figure 6c). High and low concentrations of fungicides at 7 days after treatment showed a clear separation among the control and treated (Figure 6d). The metabolites responsible for the separation can be seen in Figure 7.

Supervised MvDA was conducted to get clearer separation for all of the samples both from high and low concentrations (Figure 7). PLS-DA model resulted in a variance of R^2^ of 0.7045 and a predictive ability Q^2^ of 0.5950. The cross validation of the model using CV-ANOVA gave highly significant results (F = 10.6133, *p* < 0.001). OPLS-DA managed to separate all of the samples between the control and treated; interestingly, the model was also able to discriminate fungicides-treated samples between the systemic, represented by Pronto Plus samples, and the non-systemic, represented by Switch. However, putting all the samples together and separating between high and low levels of fungicide, resulted in a different pattern in which Folicur samples were situated close to Switch.

Loganin, secologanin, and glucose were high in Pronto Plus-treated samples while vindoline, catharanthine, 4-o-Caffeoyl quinic acid, chlorogenic acid, acetic acid, and sucrose were found to be high in the control.

Quantitative analysis of the metabolites responsible for the separation of the control and fungicides-treated samples shows that control had a higher concentration of catharanthine (ANOVA, F = 6.926, *p* = 0,001) and vindoline (ANOVA, F = 28.227, *p* < 0,001) compared to treated seedlings. In contrast, treated seedlings had higher loganin (ANOVA, F = 14.478, *p* = 0,001) and secologanin (ANOVA, F = 13.397, *p* = 0,001) (Figure 8). These results are in line with the outcome of the HPLC analysis, as described above.

The sugars identified were glucose and sucrose. Glucose (ANOVA, F = 13.415, *p* < 0,001) was significantly higher in the treated seedlings with Pronto Plus, Folicur, and Switch, whereas sucrose (ANOVA, F = 6.575, *p* < 0,001) was significantly higher in the control.

### 2.5. Endophytes Detection

As shown in the ^1^HNMR spectra (Figure 3 and Figure 4), no fungicide residues were left in the seedlings at the lower level treatments. To prove the presence or absence of endophytes in the seedlings, plant parts were put on a medium suited for endophyte growth. Unfortunately, growth of fungi was observed in neither the control nor the treated samples. It was also not observed by means of PCR using specific primers to identify the internally transcribed spacer region from (endophytic) fungi. This approach also failed to detect any fungi (Figure 9).

## 3. Discussion

The results of our study show that *C. roseus* plants, treated or not treated with fungicides, were able to synthesize alkaloids, indicating that endophytic fungi were not directly involved in alkaloids synthesis. However, endophytic fungi affect both qualitatively and quantitatively the metabolites produced in plants.

In all three fungicides treatments, Pronto Plus, Folicur, and Switch, alkaloids were produced. None of the approaches conducted (planting part of the plant organ into PDA medium, and PCR analysis) showed the presence of endophytic fungi in the samples. Endophytic fungi in seedlings might be present in concentrations that are too low for detection. Although previous studies reported a simple isolation of endophytes from *C. roseus* [3,26]. ITS1_FKYO2 had less than 10% coverage of plant DNA as it was expected to be selective for certain endophytes [28], but this did not help to detect the presence of endophytic fungi in the *C. roseus* used in the experimental samples.

The metabolomics analysis of the control and fungicide treatments showed separation between the control and treated samples. Pronto Plus and Folicur samples are systemic fungicides and the seedlings treated with these fungicides were separated clearly from control, by the presence of high levels of loganin and secologanin (Figure 5). Loganin or a precursor (loganic acid) has to be transported to the leaf epidermis, where it is converted to secologanin, the precursor for the terpenoid indole alkaloid. The increased levels of these precursors were not followed by an increased alkaloids accumulation. The non-systemic fungicide (Switch) showed similar levels of loganin and secologanin as control samples. It seems that systemic fungicides greatly affected certain secondary metabolite pathways. It was not clear if this was an effect directly on the plant or an indirect effect on endophytes. The high levels of the iridoids were either due to an effect of the biosynthesis in the iridoid-producing leaves’ cells; a block of the iridoid transport to the epidermis cells producing the alkaloids; reducing the catabolism of the iridoids; an inhibition of the alkaloid biosynthesis; or a combination of these. The alkaloid levels seem to increase in the controls over the 28 days period of the experiment, but in treated seedlings the levels seem not to be significantly increased, whereas at the same time a large increase in the iridiods’ levels is observed after treatment with the systemic fungicides. This points to a more complex regulation of the biosynthetic pathways.

The different effects of the two types of fungicides might be related to their mode of action. Systemic fungicides were absorbed in the xylem, transported to the upper part, and distributed throughout the plant. This type of fungicide showed a broad spectrum of activity against fungi, such as Ascomycetes, Basidiomycetes, and Fungi Imperfecti. The non-systemic fungicide could not be transported into the plant and acted locally only. The non-systemic fungicides had a much smaller spectrum of fungicidal activity [23,29].

Catharanthine and vindoline were higher in the controls than in the treated seedlings, a difference which is statistically significant at day 28. This is similar to the result of *C. roseus* treated with endophytic bacteria [28].

Endophytic fungi were discovered in angiosperms, gymnosperms, ferns, and mosses; interestingly most of the fungi were Ascomycetes; other fungi, such as Basidiomycetes and Deuteromycetes, were less common. It can be said that they occurred in more than 90% of all plant species studied for endophytes. Each plant was thought to harbor at least one endophytic fungi [4]. Though in seedlings, the endophytes might not be present or are not yet active. Although endophytic fungi appear to be universally associated with plants, difficulties related to their detection and monitoring have largely limited the study of their influence on plants.

## 4. Materials and Methods

### 4.1. Chemicals and Materials

Methanol for HPLC grade and analytical reagent was bought from Sigma Aldrich, GmbH (Steinheim, Germany). Ortho-phosphoric acid (H_3_PO_4_, 85 %) and disodium hydrogen phosphate (Na_2_HPO_4_) were bought from Merck (Darmstadt, Germany). Strictosidine and secologanin were from Phytoconsult (Leiden, The Netherlands); loganic acid, loganin, tabersonine, and vindoline were bought from PhytoLab (Vestenbergsgreuth, Germany); tryptamine was purchased from Aldrich Chemical (Milwaukee, WI, USA); tryptophan and ajmalicine were purchased from Sigma-Aldrich (St. Louis, MO, USA); serpentine was purchased from Roth (Karlsruhe, Germany); catharanthine, anhydrovinblastine, vinblastine, and vincristine were kind gifts from Pierre Fabre (Gaillac, France). Fungicides used were Folicur, Pronto Plus, and Switch, which have different modes of action.

### 4.2. Effect of Fungicides on the Fungus Growth

PDA (potato dextrose agar) plates were mixed with different concentrations of fungicides (Pronto Plus, Folicur, and Switch), ranging between 0.01%, 0.02%, 0.03%, 0.04%, 0.05%, 0.005%, and 0.001%. Each concentration was represented by three replicates. A mycelial plug (d = 5 mm) from a 14-day-old culture was placed on the media after being solidified and grown in the dark incubator at 27 °C for 14 days.

### 4.3. Effect of Fungicides on C. roseus Seedlings

In a preliminary experiment, fungicide concentrations used were 0.05%, 0.10%, 0.15%, 0.03%, 0.001%, and 0.005%. The concentrations used for further experiments were 0.001% Pronto Plus and Switch, as well as 0.005% Folicur. The fungicides were added to 50 mL MS (Murashige–Skoog) medium in the plastic growth box before solidification under sterile condition. Controls contained MS medium only (El-Sayed and Verpoorte 2004).

### 4.4. Seedling Germination

Seeds of *C. roseus* are surface-sterilized. The seeds were submerged in absolute ethanol for 10 s and rinsed in sterile distilled water. Then, seeds with intact testae were submerged for 20 min in 10% calcium hypochlorite, washed three times in sterile water, and germinated on MS medium. After two weeks, every ten seedlings were transferred to separate experimental plastic boxes (11 × 5 × 8 cm^2^) containing MS medium with or without fungicide; seedlings were grown under 16 h of artificial light (81 µmol m^−2^s^−1^) at 25 °C. The seedlings were harvested at the first week after treatment. The fresh weight was recorded before the seedlings were frozen in liquid nitrogen.

### 4.5. Standard Solution

The stock solutions were prepared at a concentration of 5 mg, precisely weighed, and dissolved in 1.00 mL methanol. The calibration curves were made by using six levels of the standards. Level six was prepared by diluting (1:50, *v/v*) stock solutions, which was further diluted by a factor of 2, 4, 8, 16, and 32 to get level five to one.

### 4.6. Alkaloid Extraction

The alkaloids were extracted following the slightly modified *Catharanthus* alkaloid method [30]. Freshly collected sample materials were ground in liquid nitrogen using mortar and pestle. Subsequently, the samples were lyophilized for 72 h. An amount of 50 mg of dried sample for every time point in triplicate (7, 14, 21, and 28 days) of each treatment (control, Pronto Plus, Folicur, Switch) was extracted two times by addition of 5 mL methanol, vortexed for 10 s, then sonicated for 20 min, and centrifuged at 3500 rpm for 30 min at room temperature. Then, 10 mL of the supernatant was concentrated and suspended with 0.5 mL of 1 M H_3_PO_4_, vortexed for 10 s, and transferred to a tube and centrifuged at 13,000 rpm for 10 min. After filtering with 0.20 µm PTFE membrane, the samples were ready for HPLC injection.

### 4.7. HPLC Analysis

The HPLC system was an Agilent 1200 Series, consisting of a G1310A binary pump, a G1329A auto sampler, a G1322A degasser, and a G1315D photo-diode array detector controlled by ChemStation software (Agilent v. 03.02; all from Agilent Technologies Inc., Santa Clara, CA, USA). The chromatography was conducted using a Zorbax XDB C18 column (4.6 × 250 mm, 5 µm) (Agilent Technologies, Santa Clara, CA, USA) and a guard column Eclipse XDB-C18 (4.6 × 12.5 mm, 5 µm) (Agilent Technologies, Santa Clara, CA, USA). The mobile phase consisted of 5 mM Na_2_HPO_4_ (solvent A, pH adjusted to 6 with H_3_PO_4_) and methanol (solvent B). The eluent profile was a 0–26 min linear gradient A: B from 86:14 to 14:86, 26–30 min isocratic elution with 14:86, 30–35 min linear gradient from 14:86 to 86:14, 35–37 min isocratic elution with 86:14. The injection volumes and flow rate were 30 μL and 1.5 mL/min, respectively. Each replicate was injected twice (Pan et al., 2012). UV-spectra of all peaks were collected (220–320 nm) and chromatograms were recorded at 220, 254, 280, 306, and 320 nm.

### 4.8. Extraction and ^1^HNMR Measurement

An amount of 25 mg of the freeze dried material was transferred to 2 mL Eppendorf tube and 1.5 mL of a mixture of phosphate buffer and methanol-d4 (1:1), containing 0.01% trimethylsilyl propionic acid sodium salt-d4 (TMSP, *w/w*) was added. The mixture was vortexed for one minute, ultrasonicated for thirty minutes, and centrifuged for twenty minutes at 13,000 rpm at room temperature. Aliquots were transferred to a new Eppendorf tube for second centrifugation at 13,000 rpm for five minutes, after which 800 µL of the supernatants was transferred to NMR tube for measurement (Kim et al., 2010).

The ^1^HNMR spectra were recorded using a Bruker AV 600 MHz spectrometer (Bruker, Karlsruhe, Germany) at 25 °C, and consisted of 128 scans requiring 10 min and 26 sec acquisition with following parameters: 0.16 Hz/point, pulse width of 30 (11.3 µs), and relaxation delay of 1.5 sec. Free induction decay was Fourier transformed with a line-broadening (LB) factor of 0.3 Hz. The resulting spectra were manually phased, baseline corrected, and calibrated to trimethylsilyl propionic acid sodium salt, *d_4_*(TMSP, *d_4_*), at 0.0 ppm by using XWIN NMR version 3.5 (Bruker, Karlsruhe, Germany). After processing, ^1^HNMR spectra were automatically binned by AMIX software (v.3.7, Biospin, Bruker) to include regions of equal width (δ 0.04), corresponding to the region of δ 0.04–10.00. The regions of δ 4.75–4.90 ppm and δ 3.30–3.35 ppm were excluded from the analysis because of the residual signal of water and methanol, respectively. Multivariate data analysis was performed with SIMCA-P+ software version 12.0 (Umetrics, Umea, Sweden).

### 4.9. Endophytes Detection

At each harvesting time, each plant’s organ (root, stem, and leaves) was cut into pieces and planted onto potato dextrose agar (PDA) to check the presence of endophytic fungus in the plant tissue.

To detect the presence of endophytic fungi in the seedlings, genes specific for fungi-using primers of the β-tubulin gene and internal transcribed spacer (ITS) region of the rDNA gene cluster were amplified. Total DNA was extracted from dry seedlings material using Dneasy Plant Mini Kit (QIAGEN). The amplification of the partial β-tubulin gene was carried out using β-tubulin 3-LM and β-tubulin 10-LM primers. Amplification of the ITS region of the rDNA gene cluster was performed with the primers ITS1-F_KYO2 and ITS2_KYO2, Toju et al., [31]. Total DNA of *Chaetomium* sp. was used as positive fungal control. Amplification was performed in 20 µL aliquots consisting of 2 µL 10 × PCR buffer; 0.5 µL 25 mM MgCL_2_; 0.5 µL dinucleotide triphosphates (dNTPs) containing 10 µM of each base; 1 µL of each primer; 10 µM, 0.2 µL, and 5 U/µL *Tag* polymerase (QIAGEN, Venlo, The Netherlands); and 1 µL of stock DNA. The PCR conditions were: 5 min at 95 °C, denaturation 30 s at 95 °C, annealing 1 min at 55 °C (for the ITS region gene) or 1 min at 59 °C (for the partial β-tubulin gene), and extension 1 min at 72 °C (40 cycles) and 10 min at 72 °C. The PCR products were separated in a 1.5% agarose gel for evaluation of the product size. We then sequenced the β-tubulin and ITS PCR products derived from the control and treated-seedlings, as well as from the positive fungal samples (MACROGEN, Amsterdam, The Netherlands) to identify the genes amplified.

### 4.10. Data Analysis

The experiment was conducted to determine the difference between control and fungicides treated (Pronto Plus, Folicur and Switch) at four different time points (7, 14, 21, and 28 days after treatment) in which each of the treatments at each time point was done in triplicate. In total, there were 48 data to be analyzed. Statistical analysis was performed using one-way analysis of variance (ANOVA). The results of the individual experiments are presented as the mean value ± standard deviation from the three replicates in each treatment and time point.

## 5. Conclusions

Treatment with fungicides influenced both the primary and secondary metabolism of *C. roseus.* The systemic fungicides Pronto Plus and Folicur caused an increase in loganin and secologanin levels. But catharanthine and vindoline, the main alkaloids in the studied plants, did not show any clear increase of their levels during the fungicides experiments, in contrast with the controls, where a clear increase of the alkaloid levels was observed. It seems that the fungicides do affect secondary metabolism, though there is no clear evidence yet for a role of endophytes.

## Figures and Tables

**Figure 1 molecules-28-01405-f001:**
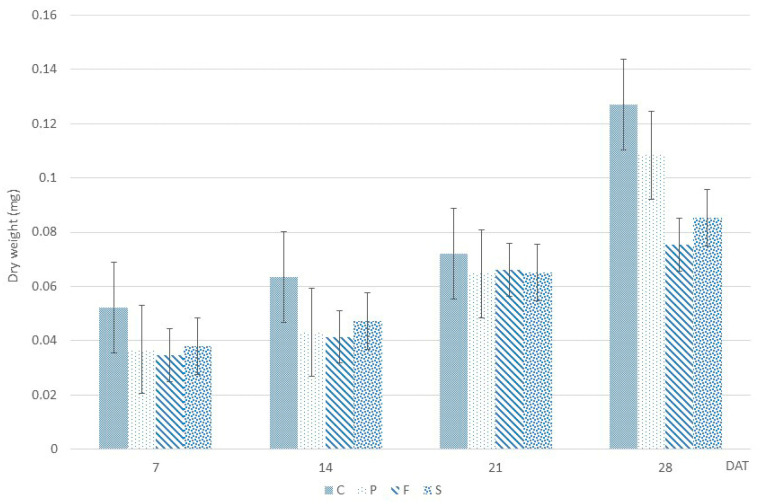
The biomass accumulation of *Catharanthus roseus* seedlings treated with different fungicides compared to untreated, harvested at different time points. The numbers correspond to harvesting time (7, 14, 21, and 28 days after treatment) and the letters to fungicides (P—Pronto Plus, F—Folicur and S—Switch) and control—C. Data present the average and standard deviation of triplicates for each time point of each condition.

**Figure 2 molecules-28-01405-f002:**
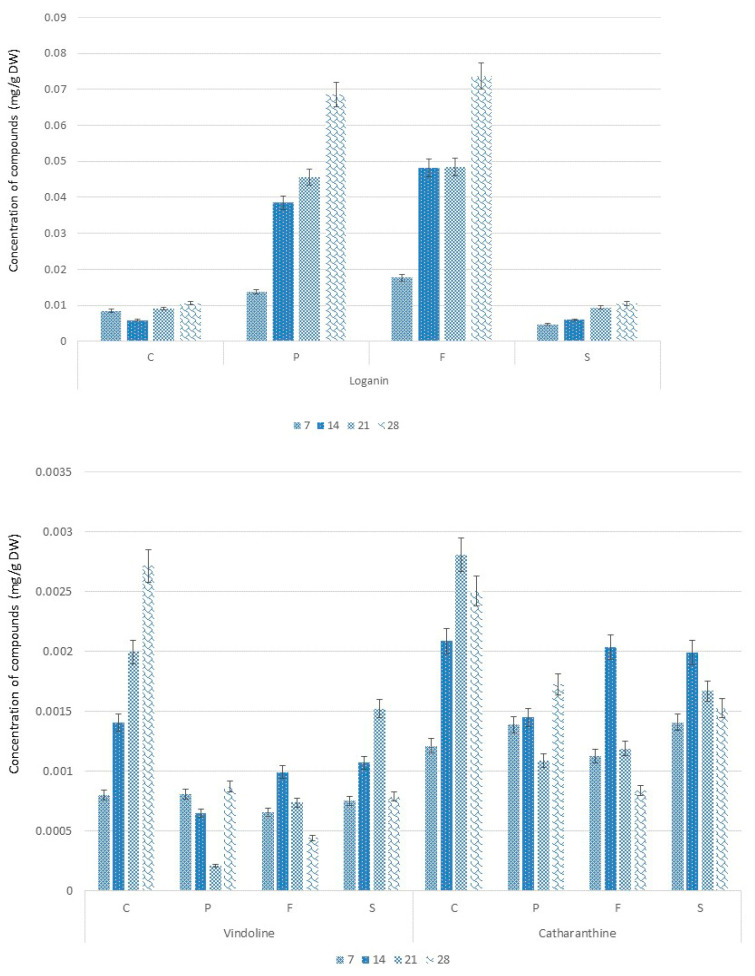
HPLC analysis of the effect of fungicide treatments (Pronto Plus (P), Folicur (F), and Switch (S)) compared to the untreated (control, C) on the concentration of loganin, catharanthine, and vindoline accumulation in *Catharanthus roseus*. The numbers correspond to harvesting time (7, 14, 21, and 28 days after treatment) and the letters to fungicides (P—Pronto Plus, F—Folicur and S—Switch) and control—C. Data present the average and standard deviation of triplicates for each time point of each condition.

**Figure 3 molecules-28-01405-f003:**
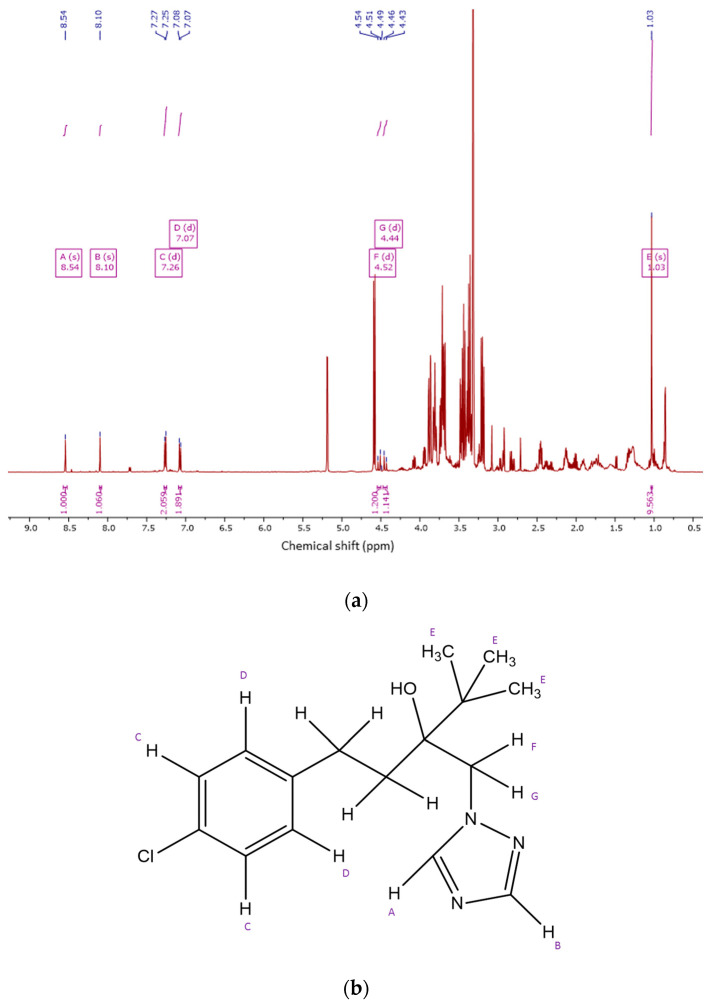
^1^HNMR spectra *C. roseus* seedlings treated with 0.03% Pronto Plus (**a**) and tebuconazole structure (**b**).

**Figure 4 molecules-28-01405-f004:**
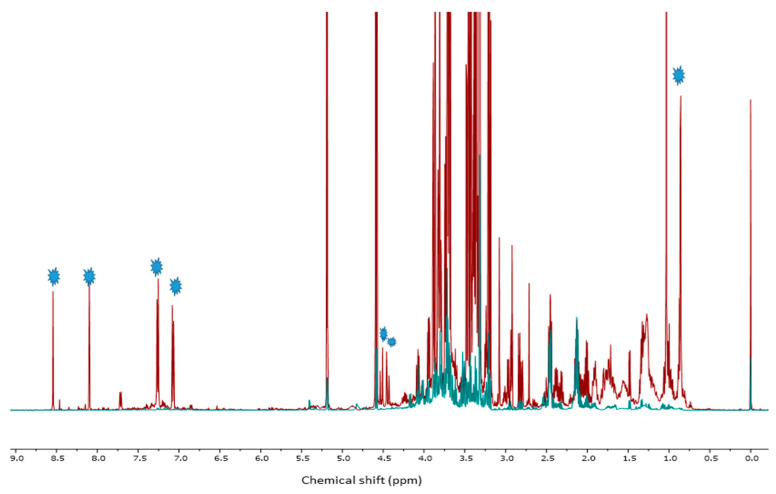
^1^HNMR spectra *C. roseus* seedlings treated with 0.03% Pronto Plus (red spectra) and 0.001% Pronto Plus (blue spectra); the blue stars indicate the fungicide residue in the samples.

**Figure 5 molecules-28-01405-f005:**
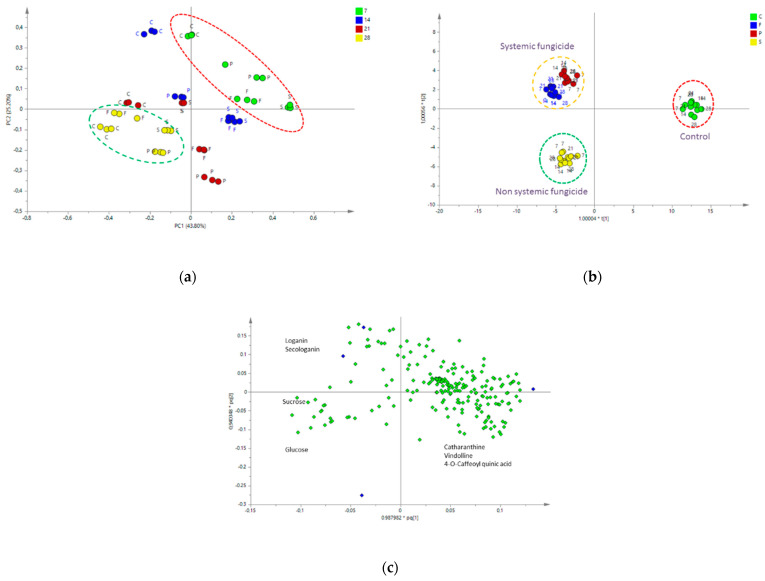
(**a**) Score scatter plot showing result of the PCA with all of the seedling samples. The numbers correspond to harvesting time (7, 14, 21, and 28 days after treatment) and the letters to fungicides (P—Pronto Plus, F—Folicur, and S—Switch) and control—C. (**b**) Score scatter plot showing result of the OPLS-DA with all of the seedling samples. (**c**) OPLS-DA loading scatter plot.

**Figure 6 molecules-28-01405-f006:**
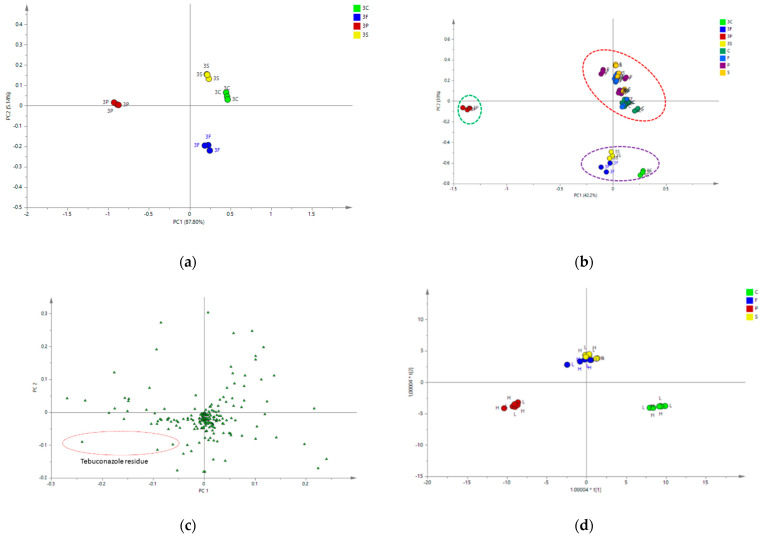
(**a**) Score scatter plot showing result of the PCA with all of the seedling samples treated with 0.03% fungicides. (**b**) Score scatter plot of all treated seedlings: 0.03% Pronto Plus, 0.03% Folicur, and 0.03% Switch; 0.001% Pronto Plus and 0.001% Folicur; 0.005% Switch. (**c**) Loadings scatter plot of all treated seedlings in b. (**d**) Score scatter plot of OPLS-DA with low and high concentration of fungicides at 7 days after harvesting. The numbers correspond to harvesting time (7, 14, 21, and 28 days after treatment) and the letters to fungicides (P—Pronto Plus, F—Folicur, and S—Switch) and control—C.

**Figure 7 molecules-28-01405-f007:**
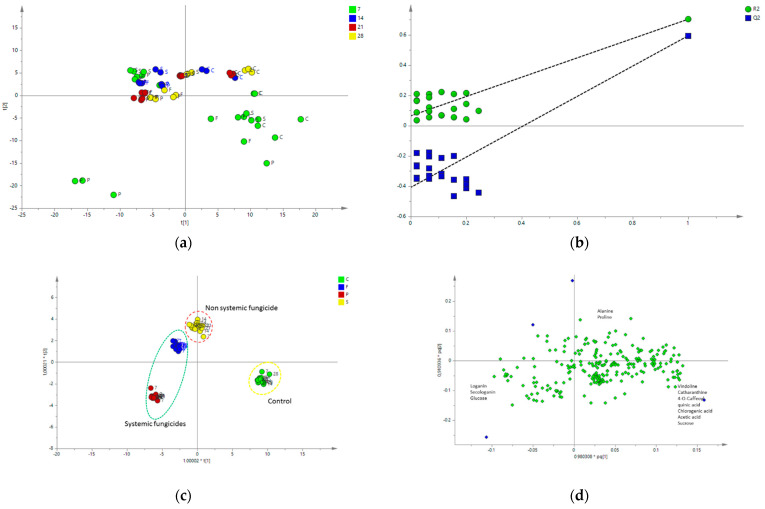
(**a**) Score scatter plot showing result of the PLS-DA with all of the seedling samples treated with 0.03% fungicides. (**b**) Permutation of the PLS-DA model. (**c**) Score scatter plot of OPLS-DA. (**d**) OPLS-DA loadings scatter plot of the all treated seedlings: 0.03% Pronto Plus, 0.03% Folicur, and 0.03% Switch; 0.001% Pronto Plus and 0.001% Folicur; 0.005% Switch. The numbers correspond to harvesting time (7, 14, 21, and 28 days after treatment) and the letters to fungicides (P—Pronto Plus, F—Folicur, and S—Switch) and control—C.

**Figure 8 molecules-28-01405-f008:**
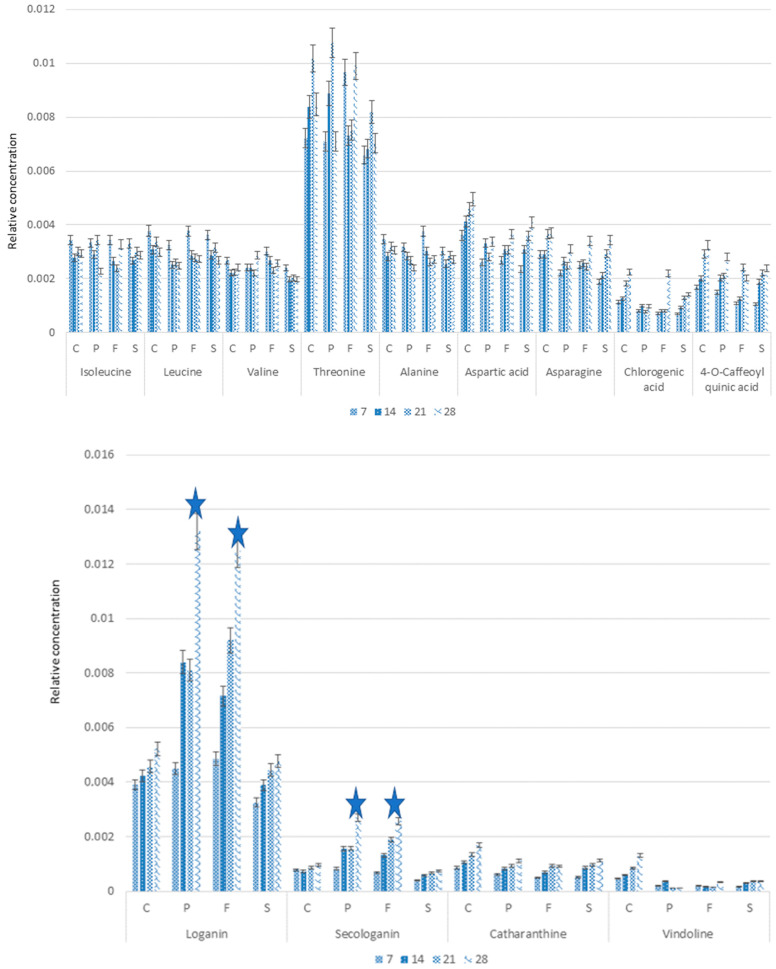
^1^HNMR-based quantification of metabolites in *C. roseus* seedlings samples after fungicides treatment in several different time points (means ± SD, *n* = 3). The numbers correspond to harvesting time (7, 14, 21, and 28 days after treatment) and the letters to fungicides (P—Pronto Plus, F—Folicur and S—Switch) and controlC. Data present the average and standard deviation of triplicates, * showing significance at *p* < 0.05.

**Figure 9 molecules-28-01405-f009:**
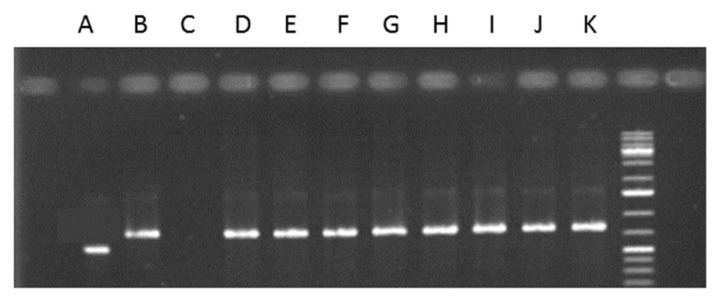
ITS_KYO2 primers used to determine the presence of fungal and plant DNA. A. Extracted endophytic fungi DNA isolated from *C. roseus* as positive control, B. Extracted DNA from *C. roseus*, C. Water as negative control, D. Control samples, E. Pronto Plus samples, F. Folicur samples, G. Switch samples, H. Control samples, I. Pronto Plus samples, J. Folicur samples, K. Switch samples. D–G = 7 days after treatment samples, H–K = 28 days after treatment samples.

**Table 1 molecules-28-01405-t001:** ^1^H chemical shifts (δ) and coupling constants (Hz) of *Catharanthus roseus* seedlings metabolites in methanol-d_4_-KH_2_PO_4_ in D_2_O at pH 6.0.

No.	Metabolite	Chemical Shifts (δ) and Coupling Constants (Hz)
1	Isoleucine	δ 0.95 (t, *J* = 7. Hz), δ 1.03 (d, *J* = 6.8)
2	Leucine	δ 0.98 (d, *J* = 6.2 Hz), δ 0.97 (d, *J* = 6.1 Hz)
3	Valine	δ 1.00 (d, *J* = 7.0 Hz), δ1.03 (d, *J* = 7.0 Hz), δ 2.28 (m)
4	Threonine	δ 1.34 (d, *J* = 6.6 Hz)
5	Alanine	δ 1.48 (d, *J* = 7.2 Hz)
6	Arginine	δ 1.67 (m), δ 1.75 (m), δ 1.92 (m)
7	Choline	δ 3.22 (s)
8	Proline	δ 4.07 (dd, *J* = 8.6 Hz, 6.4 Hz), δ 2.31 (m),
9	Glutamine	δ 2.45 (m), δ 2.13 (m)
10	Aspartic acid	δ 2.82 (dd, *J* = 16.9 Hz, 3.9 Hz), δ 2.95 (dd, *J* = 17.0 Hz, 8.2 Hz)
11	Asparagine	δ 2.82 (dd, *J* = 16.9 Hz, 8.2 Hz), δ 2.96 (dd, *J* = 16.9 Hz, 3.9 Hz)
12	Acetic acid	δ 1.94 (s)
13	Myoinositol	δ 3.47 (dd, *J* = 10.0 Hz, 2.9 Hz), δ 3.62 (t, *J* = 9.7 Hz)
14	Sucrose	δ 5.41 (d, *J* = 3.8), δ 4.17 (d, *J* =8.6)
15	Glucose	δ 5.18 (d, *J* = 3.8), δ 4.58 (d, *J* = 7.9)
16	Chlorogenic acid	δ 7.62 (d, *J* = 15.9 Hz), δ 7.14 (d, *J* = 2.1 Hz), δ 7.05 (dd, *J* = 8.4 Hz, 2.2 Hz)
17	4-o-Caffeoyl quinic acid	δ 7.67 (d, *J* = 15.9), δ 7.17 (d, *J* = 2.1), δ 7.08 (dd, *J* = 8.3, 2.0), 2.1), δ 6.44 (d, *J* = 15.9), 2.09 (m)
18	Loganin	δ 7.42 (d, *J* = 1.2), δ 5.34 (d, *J* = 4.0), δ 4.73 (d, *J* = 8.0), δ 1.08 (d, *J* = 7.2).
19	Secologanin	δ 7.44 (dd, *J* = 11.7, 0.9), δ 7.56 (d, *J* = 1.9), δ 9.65 (d, *J* = 1.4)
20	Catharanthine	δ 1.10 (t, *J* = 7.3), δ 7.36 (d, *J* = 8), δ 7.55 (d, *J* = 8)
21	Vindoline	δ 0.51 (t, *J* = 7.4), δ 2.00 (s), δ 2.66 (s), δ 5.93 (m), δ 6.22 (d, *J* = 2.3), δ 7.11 (d, *J* = 8)

## Data Availability

Available on request from authors.

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
