# Peer review of "Do Fungicides Affect Alkaloid Production in Catharanthus roseus (L.) G. Don. Seedlings?"

_molecules, 2023, doi:10.3390/molecules28031405_

Round 1

Reviewer 1 Report

Dear Authors

The MS entitled “Do Fungicides Affect Alkaloid Production in Catharanthus roseus (L.) G. Don. Seedlings?” was thoroughly reviewed. The article is technically correct however, there are some serious concerns in the formulation of MS.

General comments:

·         The title should be revised and should contain the endophyts as well.  

·         Provide the Email addresses of all the authors in authors affiliation sections (according to the molecules format).

·         Also, the references should be revised, DOI should be added where possible and modify the old with new ones.

Abstract:

·         Correct “Narcissus pseudonarcissus cv”

·         Rephrase and make this paragraph concise and more meaning full. Although, this part is not suitable here so its optional. Whether remove or considering it a specific preamble, it should be made concise.

“Fungicide treatment into Taxus is known to decrease the taxol content. In Ipomoea asarifolia, Pronto Plus and Folicur treatment coincided with the disappearance of ergot alkaloids from the plant. In Narcissus pseudonarcissus cv Carlton, mixture of fungicide application is decreased the al-18 alkaloids concentration and altered the carbohydrate metabolism. Jacobaea plants treated with Folicur 19 reduced the pyrrolizidine alkaloids content”.

·         The actual; results obtained should be added in to abstract

·         Avid the use of “we know that”. Need to be revised.

·         Particularly should be particular.

Introduction:

·         References citations should be in order (starting from 1)

·         (23; 22; 32) are not acceptable. Should be in square brackets [x] and the order should be [22, 23, 32). Also, commos are separating medium not semicolon (;).

·         Line 60. Provide the website accessed date?

·         “must be at least 300.000” is not acceptable. provide actual figure.

Results ad discussion

The results are scrambled. Need to be precise. Also, the figures, 4 and 7 are of very low quality that could not be read. The NMR is confusing.

Page 5. The NMR discussion should be more precise. The signals you are referring should be labeled with proton number and type.

Author Response

Point 1: The title should be revised and should contain the endophyts as well.  

Response 1: since we cannot prove the presence of endophyte by checking using PCR on all the samples both control and treatments thus we do not mention endophyte in the title.

Point 2:  Provide the Email addresses of all the authors in authors affiliation sections (according to the molecules format).

Response 2: yes

Point 3: Also, the references should be revised, DOI should be added where possible and modify the old with new ones.

Response 3: DOI could not provided at the moment

Point 4: Rephrase and make this paragraph concise and more meaning full. Although, this part is not suitable here so its optional. Whether remove or considering it a specific preamble, it should be made concise.

“Fungicide treatment into Taxus is known to decrease the taxol content. In Ipomoea asarifolia, Pronto Plus and Folicur treatment coincided with the disappearance of ergot alkaloids from the plant. In Narcissus pseudonarcissus cv Carlton, mixture of fungicide application is decreased the al-18 alkaloids concentration and altered the carbohydrate metabolism. Jacobaea plants treated with Folicur 19 reduced the pyrrolizidine alkaloids content”.

Point 5: The actual; results obtained should be added in to abstract

Response 5: results included at the last part of the abstract

Point 6: Avid the use of “we know that”. Need to be revised.

Response 6: already modified

Point 7: Particularly should be particular.

Response 7: yes

Point 8: References citations should be in order (starting from 1)

(23; 22; 32) are not acceptable. Should be in square brackets [x] and the order should be [22, 23, 32). Also, commos are separating medium not semicolon (;).

Response 6: already changed

Point 9: Line 60. Provide the website accessed date?

Response 9: I could not provide the accessed date since written down long time ago

Point 10: must be at least 300.000” is not acceptable. provide actual figure.

Response 10: yes

Point 11:The results are scrambled. Need to be precise. Also, the figures, 4 and 7 are of very low quality that could not be read. The NMR is confusing.

Response 11: figure already fixed

Point 12:Page 5. The NMR discussion should be more precise. The signals you are referring should be labeled with proton number and type.

Response 12: I did the numbering at figure 3 and page 5 for example

Reviewer 2 Report

The authors intended to characterize the influence of endophytic fungi on the biosynthesis of alkaloids in Catharanthus roseus and the influence on its metabolism. In general, the manuscript is clearly structured and the methods used to obtain the results are well-defined. However, several aspects need to be clarified and revisions would be appropriate.

Please consider the following points for the revision of your manuscript:

The study does not monitor the effects of individual fungicidal substances, but it shows effects of commercially available preparations. In the study, they are marked with trade names, but the registered trademark symbol is missing. Furthermore, the specification of the origin of these tested products is missing.

I recommend editing the interpretation of the statistical analysis in the result part 2.2. and 2.3. One way analysis of variance (ANOVA), used in the study, can give us information about whether the groups in the set are statistically different or not, but it will no longer tell us between which specific groups this difference is. Nevertheless, in the text commenting on the results, there are statements about a statistically significant difference (e.g. in the case of the accumulation of catharanthine) between the control sample and the treated samples. An appropriate post hoc test must be used for such a claim. I recommend using a suitable post hoc test and adding significance markers to the graphs, or at least adjusting the interpretation of the results. Also, the statement “Control had the highest dry mass compared to all treatments.” (lines 101–102) is not well formulated. Dry weight values ​​of control samples and treated samples cannot be evaluated as different if the standard deviation intervals overlap.

The abstract is not well structured. It is rather a review in which there is a minimum of information about the study, its aim, methods, etc.

The third paragraph of the discussion lacks any citation, although it also provides information that does not follow from the results of this study (e.g. lines 281–283).

I recommend an additional language editing. Even in the abstract there are some mistakes but they also appear throughout the rest of the text.

Figure formatting should be improved. Graphs on one figure are not of the same size and they are often very difficult to read.

The numbering of citations does not correspond to the location of the citation in the text, it is not sequential. There is also a mistake in the numbering in the list of references.

The term “Chaetomium sp” (lines 93, 94 and 406) is missing a point after “sp“.

Author Response

Response to Reviewer 2 Comments

Point 1: I recommend editing the interpretation of the statistical analysis in the result part 2.2. and 2.3. One way analysis of variance (ANOVA), used in the study, can give us information about whether the groups in the set are statistically different or not, but it will no longer tell us between which specific groups this difference is. Nevertheless, in the text commenting on the results, there are statements about a statistically significant difference (e.g. in the case of the accumulation of catharanthine) between the control sample and the treated samples. An appropriate post hoc test must be used for such a claim. I recommend using a suitable post hoc test and adding significance markers to the graphs, or at least adjusting the interpretation of the results. Also, the statement “Control had the highest dry mass compared to all treatments.” (lines 101–102) is not well formulated. Dry weight values ​​of control samples and treated samples cannot be evaluated as different if the standard deviation intervals overlap.

Response 1: Statistical analysis was performed using one way analysis of variance (ANOVA)

Point 2:  The abstract is not well structured. It is rather a review in which there is a minimum of information about the study, its aim, methods, etc.

Response 2: already changed a little

Point 3: The third paragraph of the discussion lacks any citation, although it also provides information that does not follow from the results of this study (e.g. lines 281–283)..

Response 3: Because it discussed the graph from the figure

Point 4: Figure formatting should be improved. Graphs on one figure are not of the same size and they are often very difficult to read.

Response 5: the figures already changed into bigger size

Point 6: The numbering of citations does not correspond to the location of the citation in the text, it is not sequential. There is also a mistake in the numbering in the list of references.

Response 6: already checked and fixed

Point 7: The term “Chaetomium sp” (lines 93, 94 and 406) is missing a point after “sp“.

Response 7: Chaetomium sp is the name of endophyte isolated from C. roseus in the study and we did not study further in this part

Reviewer 3 Report

The manuscript entitled, Do Fungicides… needs major revision before it can be accepted for publication.

1.       The authors need to explain the Figures because they are all very small. The figures should be stand-alone. Don’t put abbreviations. Also, how did you come up with the error bars? It was not explained well.

2.       What are the components of the Pronto and the Folicur?

3.       Table 2. How did you know from H-NMR  that it comes from C roseus, and not from the pesticides?

4.       Figure 4 should be explained and the figure should be improved, especially the structure.

5.       Please explain Figures 5 to 7. They are too small to see also. Please improve.

6.       How did you quantify based on HMNR? Have a parallel experiment to prove this.

7.       Lines 294 to 299, you can draw this for a clearer explanation.

8.       Did you try to check the fungus present in the plant?

Author Response

Point 1: The authors need to explain the Figures because they are all very small. The figures should be stand-alone. Don’t put abbreviations. Also, how did you come up with the error bars? It was not explained well.

Response 1: The figures are already modified. The results of the individual experiments are presented as the mean value ± standard deviation from the three replicates in each treatment and time point.

Point 2:  What are the components of the Pronto and the Folicur?

Response 2: Pronto plus contains tebuconazole and spiroxamine, whereas Folicur has tebuconazole only

Point 3: Table 2. How did you know from H-NMR  that it comes from C roseus, and not from the pesticides?

Response 3: Because from the NMR spectra we were able to distinguish between the compounds coming from the fungicides and the C. roseus itself. In the lab, we run a lot of reference compounds which plants produced and use it as in house library which help a lot during the assignment.

Point 4: Figure 4 should be explained and the figure should be improved, especially the structure..

Response 4: the figures already improved into bigger size

Point 5: Please explain Figures 5 to 7. They are too small to see also. Please improve..

Response 5: already upgrade the size

Point 6: How did you quantify based on HMNR? Have a parallel experiment to prove this.

Response 6: I did it long time ago, but we used relative quantification through peak intensity relative to methanol D4

Point 7:Did you try to check the fungus present in the plant?.

Response 7: yes, I did

Round 2

Reviewer 1 Report

Dear Authors. The MS has been modified and is OK. 

Author Response

Thank you very much for the review

Reviewer 2 Report

Unfortunately, the authors either did not respond or did not reflect some comments from the first review in the actual version of the manuscript.

The study does not monitor the effects of individual fungicidal substances, but it shows effects of commercially available preparations. In the study, they are marked with trade names, but the registered trademark symbol is missing. The composition of these preparations is not even mentioned in the study.Furthermore, the specification of the origin of these tested products is missing. It would be very appropriate to reflect on this remark and modify it in the manuscript.

With regard to author's response No. 1, it is not sufficient response to the comment about misinterpretation of the results of one way analysis of variance (ANOVA) used in the study. An appropriate post hoc test must be used to recognize and to declare between which samples is the difference statistically significant.

Further, the statement “Control had the highest dry mass compared to all treatments.” (lines 101–102) is not well formulated. Dry weight values  ​​of control samples and treated samples cannot be evaluated as different if the standard deviation intervals overlap. This comment from first review is also not reflected in the current version of the manuscript.

Author Response

Point 1. The study does not monitor the effects of individual fungicidal substances, but it shows effects of commercially available preparations. In the study, they are marked with trade names, but the registered trademark symbol is missing. The composition of these preparations is not even mentioned in the study.Furthermore, the specification of the origin of these tested products is missing. It would be very appropriate to reflect on this remark and modify it in the manuscrip

Response 1. We added the information at line 57-61.

Point 2. 

With regard to author's response No. 1, it is not sufficient response to the comment about misinterpretation of the results of one way analysis of variance (ANOVA) used in the study. An appropriate post hoc test must be used to recognize and to declare between which samples is the difference statistically significant.

Further, the statement “Control had the highest dry mass compared to all treatments.” (lines 101–102) is not well formulated. Dry weight values  ​​of control samples and treated samples cannot be evaluated as different if the standard deviation intervals overlap. This comment from first review is also not reflected in the current version of the manuscript.

Response 2. The ANOVA showed rejection to the null hypothesis which means there were no equal mean. thus we analyzed further using post hoc and the results was clearly showed that control separated from the rest of the samples and it had the highest value. Thus we written down as in the line 101-102.

Reviewer 3 Report

Please redraw Figure 4b. 

Author Response

Point 1. Please redraw Figure 4b. 

Response. The figure is the best I could give at the moment due to lisence problem